# Pharmacological Activity of Cepharanthine

**DOI:** 10.3390/molecules28135019

**Published:** 2023-06-27

**Authors:** Ke Liu, Bixia Hong, Shuqi Wang, Fuxing Lou, Yecheng You, Ruolan Hu, Amna Shafqat, Huahao Fan, Yigang Tong

**Affiliations:** 1College of Life Science and Technology, Beijing University of Chemical Technology, Beijing 100029, China; 2021201114@buct.edu.cn (K.L.); 2021400294@buct.edu.cn (B.H.); 2021201124@buct.edu.cn (S.W.); 2021201125@buct.edu.cn (F.L.); 2021410041@buct.edu.cn (Y.Y.); 2022210841@buct.edu.cn (R.H.); 2021420029@buct.edu.cn (A.S.); 2Beijing Advanced Innovation Center for Soft Matter Science and Engineering, Beijing University of Chemical Technology, Beijing 100029, China

**Keywords:** cepharanthine, pharmacological activity, alkaloids, antivirals, pharmacokinetics

## Abstract

Cepharanthine, a natural bisbenzylisoquinoline (BBIQ) alkaloid isolated from the plant *Stephania Cephalantha Hayata*, is the only bisbenzylisoquinoline alkaloid approved for human use and has been used in the clinic for more than 70 years. Cepharanthine has a variety of medicinal properties, including signaling pathway inhibitory activities, immunomodulatory activities, and antiviral activities. Recently, cepharanthine has been confirmed to greatly inhibit SARS-CoV-2 infection. Therefore, we aimed to describe the pharmacological properties and mechanisms of cepharanthine, mainly including antitumor, anti-inflammatory, anti-pathogen activities, inhibition of bone resorption, treatment of alopecia, treatment of snake bite, and other activities. At the same time, we analyzed and summarized the potential antiviral mechanism of cepharanthine and concluded that one of the most important anti-viral mechanisms of cepharanthine may be the stability of plasma membrane fluidity. Additionally, we explained its safety and bioavailability, which provides evidence for cepharanthine as a potential drug for the treatment of a variety of diseases. Finally, we further discuss the potential new clinical applications of cepharanthine and provide direction for its future development.

## 1. Introduction

As a typical member of the bisbenzylisoquinoline alkaloid family [1], cepharanthine (Figure 1) is mainly obtained by isolation and extraction from the tuberous roots of *Stephania Cepharantha Hayata* [2], with a content of approximately 19.5–33.5% in the root mass [3]. Cepharanthine is an amphiphilic [4], ether-soluble, spinning, nonphenolic, amorphous tertiary base and is usually crystallized from benzene [5].

Cepharanthine exhibits a variety of potent activities, including inhibition of the NF-κB signaling pathway [6], activation of the AMPK (adenosine 5′ monophosphate-activated protein kinase) signaling pathway [7], blockade of autophagosome–lysosome fusion [8], inhibition of the STAT3 (signal transducer and activator of transcription 3) signaling pathway [9], scavenging of free radicals and prevention of lipid peroxidation [10], and binding to heat shock proteins [11]. Moreover, it also plays a role in processes such as inhibition [12,13] or promotion [14,15] of cell proliferation and cell cycle arrest [16,17,18,19]. It also exhibits antiviral [12,20,21,22,23,24,25,26,27], antiparasitic [28,29,30,31,32,33,34,35], antimicrobial [36,37,38], bone resorption inhibition [39,40,41], antitumor [6,12,13,16,42,43], anti-inflammatory [7,44,45,46,47,48,49], and immunomodulatory properties [22,50,51,52,53]. Cepharanthine has applications in the treatment of many acute and chronic diseases [54], such as thanatophidia bite [55,56,57,58,59], radiation leukopenia [55,56,57,58,59,60,61,62,63,64,65,66,67,68], alopecia areata [69], and baldness [14,15,70,71]. It has been used for more than 40 years. No safety issues have been observed with the use of cepharanthine, and few reports of associated adverse effects have been reported [5]. A wide range of pharmacological activities of cepharanthine have been proved by a number of studies [6,12,13,16,20,21,22,23,24,25,26,27], but the mechanisms of its pharmacological activities still needs to be further explored. Although it has been reported that cepharanthine can inhibit viral protein activity [72,73], the regulation of the host by cepharanthine may be a more important part of its antiviral and antitumor activity. For example, cepharanthine inhibited the infection of HIV-1 [74,75,76], HCoV-OC43 [77], PRRSV [78], HTLV-1 [79], and the angiogenesis and growth of human oral squamous cell carcinoma OSCC cells [80] by inhibiting the NF-κB signaling pathway; inhibited the proliferation of breast cancer [81] by activation of AMPK signaling pathway; played an inhibitory role in HSV-1 infection [82], ovarian cancer [83] and bone resorption in vivo [39] by inhibiting the PI3K/Akt pathway.

The mechanism of action of cepharanthine is extensive and complex, and this article primarily presents the different activities and applications of cepharanthine, aiming to describe the mechanism of action and correlation of different pharmacological effects of cepharanthine.

## 2. Activity and Use of Cepharanthine

### 2.1. Antipathogenic Activity

Cepharanthine possesses inhibitory effects on various pathogens, such as viruses, parasites, and bacteria. Currently, it has even been approved as an anti-SARS-CoV-2 drug for clinical trials due to its significant inhibitory effect against SARS-CoV-2 (https://www.newsfilecorp.com/release/105841 (accessed on 20 April 2023)).

#### 2.1.1. Antiviral Effect

Cepharanthine has antiviral effects on a variety of viruses, including SARS-CoV-2 [20,21,24,27,72,73,84,85,86,87,88,89,90,91], SARS-CoV [73,92], MERS-CoV [92], HCoV-OC43 [77], SARS-CoV-2-like GX_P2V [21,93], porcine epidemic diarrhea virus (PEDV) [94], and porcine acute diarrhea syndrome coronavirus (SADS-CoV) [95], as well as other viruses, including human immunodeficiency virus type 1 (HIV-1) [26,74,75,76,96], herpes simplex virus type 1 (HSV-1) [82,97], Ebola virus [20], Zika virus [20], porcine reproductive and respiratory syndrome virus (PRRSV) [78], porcine circovirus type 2 (PCV2) [98], human T-lymphotropic virus type 1 (HTLV-1) [79], and coxsackie virus B3 (CV-B3) [99], making it a potential broad-spectrum therapeutic agent for the treatment of viral diseases. In addition, membrane fluidity is important when the most enveloped viruses enter host cells, and cepharanthine stabilizes plasma membrane fluidity (Table 1).

SARS-CoV-2;

The most recent heavyweight study on cepharanthine is its proven significant inhibitory effect on SARS-CoV-2 in vitro (molecular level, cellular level) as well as in vivo (experimental animal level). In vitro data showed that cepharanthine has a better anti-SARS-CoV-2 effect than remdesivir (EC50 = 0.1 μM vs. EC50 = 0.72 μM) [100], showing vital research value in the late pandemic period. Fan et al. [93] first discovered the potential inhibitory activity of cepharanthine against SARS-CoV-2 infection, and cepharanthine was the most effective drug against GX_P2V (an alternative model for SARS-CoV-2) activity among 2711 approved drugs, the EC50 values of cepharanthine on Vero E6 cells were 0.98 μM, the CC50 values of cepharanthine on Vero E6 cells were 39.30 μM. Compared with the control group, 10 μM cepharanthine showed 12,459-fold, 2.18-fold, and 1618-fold inhibition against GX_P2V infection in the full-time, entry, and post-entry groups, respectively, implying that cepharanthine inhibited the virus at all stages of infection. Subsequently, Ohashi et al. [72] found that cepharanthine and nelfinavir (a protease inhibitor) had a synergistic effect on inhibiting SARS-CoV-2 infection, and the combination of the two drugs could reduce the viral RNA level to 0.068% of the untreated control. This result was confirmed by the study that oral administration of nelfinavir and intravenous administration of cepharanthine reduced the clearance time of SARS-CoV-2 by 1.23 days compared with nelfinavir alone in patients with COVID-19 by a mathematical prediction model [72]. Meanwhile, He CL et al. [101] found that cepharanthine was the most effective of 188 natural compounds against SARS-CoV-2 infection. Using the SARS-CoV-2 S (G614) pseudovirus, it was found that the EC50 values of cepharanthine on 293T-ACE2, Calu3 and A549-ACE2 cells were 0.351 μM, 0.759 μM and 0.911 μM, respectively. Cell–cell fusion experiments showed that 5 μM cepharanthine effectively inhibited SARS-CoV-2 S-mediated membrane fusion in 293T-ACE2 cells, reducing the fusion rate by about 90%.

At the same time, the inhibition against SARS-CoV-2 variants and in vivo activity of cepharanthine were further confirmed. Shaojun Zhang et al. [20] found that cepharanthine was at least six-fold more active in inhibiting SARS-CoV-2 B.1.351 than the wild type. The EC50 values of SARS-CoV-2 S-N501Y. V1 (B.1.1.7) and N501Y. V2 (B.1.351) pseudoviruses in 293T-ACE2 cells were 0.047 μM and 0.296 μM, respectively. The data on the hACE2 mouse model verified that the viral load was significantly lower in the cepharanthine-treated group than in the control group and that both lung injury and inflammation caused by SARS-CoV-2 infection were less severe [20].

The antiviral activity of cepharanthine was always correlated with its anti-inflammatory activity. Using transcriptomic analysis, Li et al. [21] found that cepharanthine effectively reversed most dysregulated genes and pathways in GX_P2V (an alternative SARS-CoV-2 model)-infected cells, including the endoplasmic reticulum stress/unfolded protein response and heat shock factor 1 (HSF1)-mediated heat shock response, thereby exerting an anti-coronavirus effect. It has also been shown that cepharanthine improves cellular resistance to SARS-CoV-2 by inhibiting the NPC1 protein [84,102]. Cepharanthine can be transported to the lysosome after entering cells, where it physically interacts with the NPC1 protein and inhibits the NPC1 protein, resulting in lysosomal cholesterol accumulation and an increase in intralysosomal pH, and cepharanthine disrupts cellular/lysosome lipid homeostasis, stimulating the activity exhibited by cepharanthine against SARS-CoV-2 [84,102] (Figure 2).

In addition, cepharanthine is also thought to exert its antiviral activity by directly acting on viruses. First, molecular docking experiments demonstrated that cepharanthine could bind to the RBD and, thus, block the interaction of the virus with ACE2 [72] (Figure 2). Molecular dynamics (MD) simulation showed that cepharanthine could affect the binding affinity and free energy binding value of ACE2 to the S protein of SARS-CoV-2 wild type, H49Y, T573I, and D614G mutants [85]. Cell experiments also showed that cepharanthine had an effective antiviral effect by interfering with the binding of SARS-CoV-2 S protein to Vero E6/TMPRSS2 cells [72]. Furthermore, it has been suggested that virus replication may be inhibited by cepharanthine on targeting the NSP12-NSP7 interface, the NSP12-NSP8 interface of SARS-CoV-2, and the NSP12-NSP8 interface of SARS-CoV [73].

In conclusion, cepharanthine can potentially reduce inflammatory responses both in vivo and in vitro and has great application prospects. It may be an effective drug for treating SARS-CoV-2-induced COVID-19, but more in-depth mechanistic studies and clinical effect evaluations are still needed.

The remaining human coronaviruses;

SARS-CoV-2 is the seventh coronavirus known to infect humans, and the other six are SARS-CoV, MERS-CoV, HCoV-229E, HCoV-OC43, HCoV-NL63, and HCoV-HKU1. In addition to SARS-CoV-2, cepharanthine also inhibits SARS-CoV, MERS-CoV, and HCoV-OC43 infection. Zhang et al. found that the EC50 of cepharanthine against SARS-CoV was 6.0 μg/mL, and 10 μg/mL cepharanthine could completely inhibit SARS-CoV viral cytopathic effect [99]. Ruan Z et al. [73] also found that cepharanthine bound well to RdRp of SARS-CoV in the crystal structure, implying its potential inhibitory activity against SARS-CoV replication. Chen CZ et al. [92] performed high-throughput screening of the approved drug library for the assay using SARS-S and MERS-S pseudoviruses, and they identified cepharanthine as an S protein-mediated inhibitor that blocks entry into cells. Meanwhile, Huang et al. [94] discovered that the EC50 of cepharanthine against SARS-CoV and MERS-CoV S protein pseudoviruses was 0.0417 μM and 0.140 μM, respectively, indicating that cepharanthine could effectively inhibit SARS-CoV and MERS-CoV infection in vitro. Furthermore, Dong Eon Kim et al. [77] used HCoV-OC43 as an alternative model for SARS-CoV and MERS-CoV and found that cepharanthine had significant dose-dependent anti-HCoV-OC43 activity and inhibited HCoV-OC43 replication. The IC50 value for the HCoV-OC43 virus was 0.83 µM in MRC-5 cells, and preincubation of the virus with the drug before infection was found to be effective in increasing cell survival.

Porcine coronavirus;

In addition to human coronaviruses, cepharanthine also has potential therapeutic activity against porcine coronaviruses. Wang et al. demonstrated that the EC50 of cepharanthine against porcine epidemic diarrhea virus (PEDV) was 2.53 μM, and 11.1 mg/kg cepharanthine effectively reduced the PEDV load in piglets, attenuated histopathological changes, and reduced PEDV damage to the piglets’ small intestine [94]. At the same time, a screening of 3523 drugs by Huh 7 cells infected with porcine acute diarrhea syndrome coronavirus (SADS-CoV) showed that cepharanthine could also effectively inhibit SADS-CoV infection, and its EC50 against SADS-CoV in Huh 7 cells was 0.79 μM [95].

The antiviral activity of cepharanthine seems to be broad spectrum, and it has good inhibitory activity against human immunodeficiency virus type 1 (HIV-1), Ebola virus, Zika virus, herpes simplex virus type 1 (HSV-1), porcine reproductive and respiratory syndrome virus (PRRSV), porcine circovirus type 2 (PCV2), and other viruses.

Human immunodeficiency virus type 1 (HIV-1);

Some studies have shown that cepharanthine inhibits human immunodeficiency virus type 1 (HIV-1) with an IC50 value of 0.026 µM [75]. Okamoto M et al. [96] demonstrated that the combination of cepharanthine with K-12 synergistically inhibited HIV-1 production in U1 cells (a premonocytic cell line chronically infected with the virus). The inhibitory activity of cepharanthine against HIV-1 was related to its inhibitory activities on NF-κB, tumor necrosis factor-α (TNF-α) and phorbol myristate acetate (PMA) and its ability to stabilize the fluidity of the plasma membrane [74,75,76]. M Baba et al. [76] showed that cepharanthine effectively inhibited HIV-1 replication by depressing the expression of NF-κB in U1 cells without affecting their viability and proliferation. Moreover, cepharanthine dose-dependently inhibits HIV-1 replication in TNF-α- and PMA-stimulated U1 cells but not in ACH-2 cells [75]. In addition, cepharanthine stabilizes plasma membrane fluidity, which is necessary for most enveloped viruses to enter host cells. It has also been shown that cepharanthine inhibits viral entry by stabilizing the mobility of the plasma membrane and reducing HIV-1 envelope-dependent intercellular fusion and extracellular infection to inhibit HIV-1 [74].

Ebola and Zika Viruses;

Furthermore, cepharanthine was also able to inhibit Ebola and Zika viruses. Zhang S et al. [20] analyzed the vRNA-host protein interactions with Ebola and Zika viruses by ChIRP-MS. It was demonstrated that cepharanthine has great anti-EBOV Δ VP30-GFP and anti-ZIKV (MR766) viral activities in vitro with IC50 values of 0.42 μM and 2.19 μM, respectively, and there is still a lack of more detailed and in-depth systematic studies.

Herpes Simplex Virus type 1 (HSV-1);

Multiple studies have reported the inhibitory activity of cepharanthine against HSV-1 [25,82,99]. Cepharanthine at 12.5–25.0 μg/mL showed high anti-HSV activity in vitro [97,99]. There are also studies showing that HSV inhibited cells with a TC50 of 5.4 μg/mL and IC50 of 0.835 μg/mL, and the proliferation of HSV-1 in cells was effectively inhibited by cepharanthine treatment for 16 h [25,97]. Moreover, cepharanthine can exert its anti-HSV-1 activity by targeting the STING/TBK1/P62 signaling pathway [25] and the PI3K/Akt and p38 MAPK signaling pathways [82]. Cepharanthine promoted STING, TBK1, P62 phosphorylation, and LC3II expression by directly targeting the STING/TBK1/P62 signaling pathway without inducing interferon production, which promotes cellular autophagy to inhibit HSV-1 [25]. It has also been found that cepharanthine inhibits the PI3K/Akt and p38 MAPK signaling pathways, which block the cell cycle in the G2/M phase and induce apoptosis. Thus, all these activities may indirectly reduce HSV-1 infection and replication [82].

Porcine Reproductive and Respiratory Syndrome Virus (PRRSV);

Yang C et al. [78] determined that cepharanthine was one of the most effective inhibitors of 623 small molecules against porcine reproductive and respiratory syndrome virus (PRRSV) infection at both the RNA and protein levels. Moreover, treatment with 10 µM cepharanthine can further reduce TCID50 by 5.6-fold and attenuate the cytopathic effects. The inhibition of cepharanthine against PRRSV possibly occurs through an overall downregulation of the expression of PRRSV infection mediators, including integrin β1, integrin β3, ILK, RACK1, and PKCα, and ultimately inhibits the NF-κB signaling pathway.

Porcine Circovirus type 2 (PCV2);

Some studies have demonstrated that 0.003, 0.0015, and 0.00075 mg/mL cepharanthine can inhibit porcine circovirus type 2 (PCV2) from infecting PK-15 cells in a dose-dependent manner. In addition, apoptosis rates were significantly lower in all treatment groups than in the PCV2-infected group, with lysozyme 3 and Bax expression upregulated, and Bcl-2 expression downregulated. The mitochondrial apoptosis induced by PCV2 was alleviated by cepharanthine [98].

Human T-lymphotropic Lymphotropic Virus type 1 (HTLV-1);

As an NF-κB inhibitor, cepharanthine also synergistically inhibited abnormal cell proliferation induced by infection of human T-lymphotropic lymphotropic virus type 1 (HTLV-1) in combination with a tetramethylnaphthalene derivative, a selective inhibitor of adult T-cell leukemia (ATL) cells [79].

Coxsackie virus B3 (CV-B3).

Cepharanthine also exhibited high antiviral activity against coxsackie virus B3 (CV-B3) at concentrations ranging from 12.5 to 25.0 μg/mL [99].

#### 2.1.2. Anti Parasitic Activity

The damage resulting from parasitic diseases is still a common public health problem, and the inhibitory activity of cepharanthine against a variety of parasites has important clinical significance. Cepharanthine is effective against *Plasmodium* and *Trypanosoma cruzi*. It mainly targets *Plasmodium falciparum* strain W2 [28,31,33], D-6 [34], FCM29 [35], 3D7 [35], K1 [35], and *Trypanosoma cruzi* strain Y [103].

An in vitro antimalarial activity assay revealed that cepharanthine had an IC50 of 0.2 μM [31] and 0.61 μM [33] against *Plasmodium falciparum* W2, 3059 nM for FCM29, 2276 nM for 3D7, and 1803 nM for K1 [35], a good inhibitory effect at lethal concentrations of less than 100 pM [103] and did not mediate a cytotoxic response [32]. Meanwhile, the common use of cepharanthine with chloroquine (CQ), benfluralin (LUM), atovaquone (ATO), piperaquine (PPQ), and monodesethylamino diquine (MdAQ) increased the antiplasmodial activity and inhibited the antiplasmodial activity when combined with dihydroartemisinin (DHA) and mefloquine (MQ) [104]. Treatment with 200 nM cepharanthine alone was able to increase the inhibitory effect of chloroquine on *Plasmodium vivax* by approximately 15-fold [30]. In addition, the common use of chloroquine and amodiaquine (AQ) was also shown to increase the survival of *Plasmodium vivax*-infected mice and prolong the recurrence of the parasite [104].

Cepharanthine also has good antiplasmodial activity in vivo, and in a mouse model of *Plasmodium berghei* hematosis, symptoms were reduced by 47% with intraperitoneal injection of 10 mg/kg cepharanthine and by 50% with oral administration [33].

#### 2.1.3. Antibacterial Activity

The inhibitory activity of cepharanthine against *Mycobacterium leprae* and the therapeutic effect of cepharanthine against leprosy caused by *Mycobacterium leprae* were verified. Cepharanthine is more effective in treating tuberculosis and leprosy in a guinea pig model [105]. Sato S [38] went through clinical trials to demonstrate that cepharanthine can also treat and prevent leprosy. Out of 290 leprosy patients, cepharanthine completely cured 21 cases, significantly cured 139, and partially cured 115.

Moreover, cepharanthine had an antiproliferative effect on drug-resistant *Staphylococcus aureus* by enhancing the binding of methylglyoxal bis (cyclopentylamidinohydrazone) (MGBCP) to the bacteria and inhibiting their macromolecular synthesis [36]. For example, the common use of cepharanthine and MGBCP inhibited the proliferation of methicillin- and gentamicin-resistant *Staphylococcus aureus*.

### 2.2. Antitumor Activity

The growth of tumors is inhibited in many ways. Cepharanthine can contribute to inhibition of tumor proliferation [17,106,107], improving tumor sensitivity [7,43,108,109], and inhibiting tumor cell metastasis [110,111,112,113]. In addition, cepharanthine can reduce radiotherapy and chemotherapy toxicity [64,66,114,115], and enhance the body’s immune activity [50] to further exert antitumor effects (Figure 3).

#### 2.2.1. Inhibitors of Apoptosis and Autophagy

Cepharanthine can directly act on tumor cells to inhibit their growth [17,106]. Direct treatment with cepharanthine significantly reduced tumor growth in mice with Ehrlich ascites tumors [106]. Furthermore, similar results were also found in primary exudative lymphoma (PEL), where 1–10 μg/mL cepharanthine significantly inhibited the proliferation of PEL cells [12].

Induction of apoptosis;

Apoptosis can induce tumor cell death and prevent tumor cell growth in response to various stimuli [116]. Additionally, cepharanthine is thought to be involved in various apoptosis-inducing pathways to exert antitumor effects [19,107,117], including activation of caspases [7] (caspase-3 [6,118,119,120] and caspase-9 [6]), induction of reactive oxygen species production [120], and regulation of amino acid metabolism [13]. Cepharanthine is known to induce reactive oxygen species production to achieve anti-tumor effects in myeloma cells [17] and mouse lymphoma cells [121], causing upregulation of p21Waf1/Cip1 [18] and Bax [120], and downregulation of cyclin A and Bcl-2 [18]. Transcriptomic data analysis from in vivo experiments suggests that cepharanthine can modulate amino acid metabolism pathway and downregulate various metabolite levels to further induce apoptosis of hepatocellular carcinoma cells (HCCs) [13].

The ability of cepharanthine to regulate apoptosis is closely associated with a variety of factors. First, the ability does not appear to be simply proportional to concentration. Cepharanthine has a particular effect on promoting malignant glioma cells proliferation at concentrations of 1–10 μg/mL and inhibiting malignant glioma cells growth at concentrations more than 15 μg/mL. Second, cepharanthine exerted a better ability to induce apoptosis when combined with other drugs. When cepharanthine and onconase (onc) were used alone the cells could still proliferate, but when used in combination, the growth of cells was completely inhibited and the frequency of apoptosis was increased, including human promyelocytic leukemia HL-60 cells, human histiocytic lymphoma U937 cells, multiple myeloma RPMI-8228 cells, prostate cancer DU 145 cells, and prostate cancer LNCaP cells [122]. The same situation was observed in malignant glioma cells, and the use in combination of cepharanthine with nimustine hydrochloride (ACNU) has a greater apoptotic effect than use alone of cepharanthine [123]. Additionally, the increased effect of synergistic inhibition of tumor cells may be related to the fact that onc and ACNU target the same signaling pathway with cepharanthine [122].

Induction of autophagy;

mTOR kinase is a critical molecule in the induction of autophagy. The Akt signaling pathway can activate mTOR to inhibit autophagy. Cepharanthine can induce autophagy and apoptosis in cancer cells by regulating the Akt/mTOR or AMPK/mTOR signaling pathways [42]. mTOR kinase can inhibit autophagy and apoptosis, and AKT can induce changes in cell cycle distribution in regulating cell proliferation, angiogenesis, migration, and invasion. Phosphorylated AKT (Ser473) and phosphorylated mTOR were found to be significantly reduced in two breast cancer cell lines treated with cepharanthine, as were a series of downstream indicators of mTOR [81]. Cepharanthine can also induce autophagy and the death of apoptosis-resistant cells by activating AMPK [124]. In autophagy-resistant fibroblasts and autophagic cells in cancer, cepharanthine can exert anticancer effects by directly activating AMPK to induce death [125]. Autophagy-related molecules, such as light chain 3 (LC3), p38, and phosphorylated p38 in A549 cells, were also upregulated by cepharanthine, and autophagy was regulated by activating the p38 signaling pathway to prevent lung cancer [126]. It has been shown that cepharanthine may block autophagosome–lysosome fusion [8,127], inhibit the maturation of lysosomal histone B and histone D [8], and suppress autophagy and mitotic phagocytosis [127], thereby inhibiting autophagy in non-small cell lung cancer (NSCLC) cells [8]. Currently, the exact molecular mechanism by which cepharanthine inhibits autophagy by blocking autophagosome–lysosome fusion remains unclear [42].

#### 2.2.2. Cell Cycle Arrest and Inhibition of Angiogenesis

Leading to cell cycle arrest;

In addition to inducing apoptosis, cepharanthine can inhibit tumor growth by arresting the cell cycle. Cepharanthine can induce G1/S phase arrest [118,119] and DNA breakage [119], and inhibit the growth of a variety of cancer cells. These cells include the human adenosquamous cell carcinoma cell line (KMC-2) [119], human osteosarcoma cell line (SaOS2) [9], and ovarian cancer cell lines CaOV-3 and OVCAR-3 [19]. Cepharanthine likely arrests the cell cycle by inducing the expression of cyclin-dependent kinase (CDK) inhibitors [17], thereby inhibiting the STAT3 signaling pathway [9]. In addition, DNA breaks during cell proliferation by blocking the G1 phase [118,119], S phase [19], and G1 to S phase transition process [128], which induces apoptosis [119]. The combined action of cepharanthine and radiotherapy in treating oral squamous carcinoma cells leads to an increase in the sub-G1 peak [118], which causes cell cycle arrest. Moreover, cepharanthine significantly reduced the volume and weight of osteosarcoma in nude mice [9], reflecting its promising application as an anticancer drug.

Inhibition of angiogenesis;

Tumorigenesis and metastasis are associated with the process of neoangiogenesis [129], and cepharanthine can inhibit tumor angiogenesis, which in turn has an inhibitory effect on tumor proliferation [130]. Cholesterol transport is an essential and feasible drug target for anti-angiogenesis. As an inhibitor of cholesterol transport, cepharanthine can inhibit the endo-lysosomal transport of free cholesterol and LDL in endothelial cells [102]. Cepharanthine was found to inhibit angiogenesis and growth of human oral squamous cell carcinoma OSCC cells by inhibiting the expression of VEGF and IL-8, and blocking NF-κB activity [80]. Therefore, cepharanthine may be a potential antiangiogenic and antitumor drug.

#### 2.2.3. Overcoming Multidrug Resistance and Increasing Tumor Cell Sensitivity

Cepharanthine was also found to effectively reverse anticancer drug resistance [43] and exert antitumor effects.

P-glycoprotein (P-gp) is a drug efflux pump that plays a vital role in chemotherapy-induced multidrug resistance [108]. Cepharanthine acts as a modifier of P-gp-mediated multidrug resistance (MDR) in vitro [109], and its activity in reversing MDR is closely linked to P-gp. With a strong affinity for P-glycoprotein and the ability to inhibit P-glycoprotein [2], cepharanthine plays a crucial role in reversing multidrug resistance (MDR) by inhibiting the drug efflux pump [7], reducing drug efflux, and increasing drug accumulation [131], which can reverse drug resistance due to P-glycoprotein overexpression [132]. Cepharanthine, an MDR reversal agent [133], enhances the killing effect of perphenazine on SH-1, TH, and TE-1 cells, but P-gp is not significantly expressed in all three cell lines [134]. It has been found that in HL-60 cells with adriamycin multidrug-resistance, which is caused by the high expression of multidrug resistance-associated protein MRP [135], pergolide was able to alter the level of drug accumulation by reducing drug efflux [131]. In addition, combining pergolide with other P-gp modulators, such as cyclosporine A, enhanced the inhibition of cellular efflux of anticancer drugs through P-gp [136]. The effect of cepharanthine in overcoming multidrug resistance was achieved by inhibiting the photolabeling of P-glycoprotein [137,138] and inhibiting the photoaffinity labeling of P-glycoprotein with azidopine, thereby enhancing the accumulation of anticancer agents in multidrug-resistant cells [139]. The above discussion suggests that cepharanthine may overcome cellular multidrug resistance by affecting p-glycoprotein.

Cepharanthine can enhance the activity of adriamycin against multidrug-resistant cells [135]. Cepharanthine (1 mg/mL) can eliminate the degradation of ADR in resistant cells [140] and promote the accumulation of adriamycin in resistant cells [141]. Furthermore, it also improves the drug sensitivity of ADR against ADR-resistant tumors [142], and the combination with ADR can enhance the antitumor activity of ADR, which has beneficial effects in cancer patients [43]. Cepharanthine was able to reverse adriamycin resistance and enhance its antitumor activity in a dose-dependent manner in mouse P388 leukemia cells (P388/R cells) [143]. Drug-resistant cells treated with cepharanthine enhanced the blocking effect induced by adriamycin in cells [144]. Kisara S et al. [144] showed that cepharanthine significantly enhanced the sensitivity of adriamycin at the cellular level and the inhibition of thymidine nucleoside incorporation into the DNA of drug-resistant cells. This achieves enhanced sensitivity by altering the ability to induce DNA damage in cells and adriamycin sensitivity by altering the ability to induce cellular DNA damage. Cepharanthine increased intracytoplasmic Ca^2+^ concentrations [145], and after treatment with cepharanthine and ADR, the accumulation of ADR was increased. This could be due to increased Ca^2+^ influx and the increase in ADR accumulation, or may also be related to the interaction between H^+^ and HCO_3_^−^, with cepharanthine increasing ADR accumulation by affecting H^+^ flux [146]. In a mouse model of Ehrlich ascites tumors, combined intratumoral treatment with cepharanthine and adriamycin significantly reduced tumor growth compared with that in mice treated with adriamycin alone [106]. Cepharanthine is also known to overcome multidrug resistance by interfering with membrane function through binding to phosphatidylserine on the plasma membrane [147]. The effect of cepharanthine on multidrug-resistant cells was found to alter the sensitivity of resistant cells to drugs and enhance the effect of antitumor drugs mainly through its effect on the expression level of multidrug-resistant proteins and the efflux pump.

Multidrug resistance protein 7 (MRP7, or ABCC10) is a protein that transports chemotherapeutic drugs [16] and plays a vital role in multidrug resistance in cells. Thus, MRP7 is an important target for the action of cepharanthine [148]. Cepharanthine was able to increase the sensitivity of A549/GR cells to gemcitabine [148], reduce the resistance to paclitaxel of a paclitaxel-resistant cell line (KK47/TX30) isolated from a human bladder cancer cell line (KK47/WT) [149], increase the resistance to HC-2-6 and HC-7-5 cells by Flexeril and vincristine (VCR), and overcome the vincristine of the multidrug-resistant variant HC-7-5/VCR [150]. The combination of genistein with buthionine sulfoximine (BSO) helped to kill drug-resistant tumor cells [135]. In addition, the combination with doxorubicin (DTX) increased apoptosis in ovarian cancer cells of the multidrug-resistant MDR phenotype in vitro compared with doxorubicin alone, with an antitumor effect in vivo that was two-fold greater [151].

Cepharanthine not only enhances the sensitivity of chemotherapeutic agents but also enhances the sensitizing effect of radiation. In oral cancer treatment, cepharanthine in combination with radiotherapy can increase radiosensitivity [152]. Furthermore, in oral squamous carcinoma cells, cepharanthine strongly inhibited DNA double-strand breaks (DSBs) caused by radiation and enhanced the tumor radiation response [118]. Fang ZH et al. [153] found that cepharanthine may increase the radiosensitivity of HeLa cells in vitro and in vivo by reducing STAT3, Bcl-2, c-Myc, and COX-2 expression, thus delaying tumor growth.

Cepharanthine can also alter the temperature sensitivity of tumor cells and enhance the antitumor effect of heat therapy [154]. Wang Y et al. [155] explained that cepharanthine, as a heat sensitizer, enhanced the temperature sensitivity of mouse fibrosarcoma Fsa II in vitro and in vivo by increasing tumor cell apoptosis. This effect may be due to the effect of cepharanthine binding to heat shock proteins. Cepharanthine interacts with the intermediate structural domain of Hsp90α [11] and reverses the HSF1-mediated heat shock response [21]. These results provide a theoretical rationale for using cepharanthine as a heat sensitizer to enhance the temperature sensitivity of tumor cells.

#### 2.2.4. Inhibition of Cell Migration

Metastasis is a major contributor to cancer mortality; therefore, inhibition of tumor metastasis is an important approach to treating cancers [156]. Cepharanthine acts as a biological response modifier that enhances leukocyte natural killer (NK) activity to exert antitumor and antimetastatic effects [110]. In nude mice injected intravenously with human colon cancer cells, cepharanthine alone also significantly inhibited cancer metastasis [111], and intravitreal injection of cepharanthine and staphylococcal enterotoxin in transplanted rat osteosarcoma reduced lung metastasis [112]. It has been suggested that cepharanthine may exert its anti-metastatic effects through the inhibition of ICAM-1 and MMP-2 to treat metastatic cholangiocarcinoma CCA [113].

Macrophages play a crucial role in host defense, and cepharanthine exerts its antitumor or antimetastatic effects by regulating the expression of host defense mechanisms, which include the regulation of macrophages, T lymphocytes and NK cells [157]. In a study of the antitumor activity of Lewis lung cancer (3LL) primary tumors and their lung metastases, cepharanthine was found to be strongly associated with regulating the expression of host defense mechanisms [157]. Intravenous administration of cepharanthine and the antitumor drug OK-432 resulted in a more substantial tumor-killing effect of AM in rat alveolar macrophages, which contributed to the reduction in lung metastases [158]. Alternatively, a possible mechanism for inhibiting lung metastasis by cepharanthine may be its inhibitory effect on tumor metastasis by activating macrophages and inhibiting the hepatic microsomal drug metabolism system [159].

The inhibition of cell metastasis by cepharanthine reflects the significant role that cepharanthine plays in regulating host defense mechanisms and its ability to inhibit tumor metastasis.

#### 2.2.5. Reduction in Chemoradiotherapy Damage

Radiotherapy and chemotherapy, the two primary forms of cancer treatment, inevitably have specific side effects [160]. Cepharanthine has been shown to reduce the toxic side effects of radiation and chemotherapy [114,115] and restore leukopenia [62,63,64,65,66,67,68,161]. It also affects thrombocytopenia [65] and dry mouth [162,163] after radiation and chemotherapy.

There is experimental evidence that intravenous cepharanthine can prevent toxicity in acute or advanced prostate cancer caused by radiotherapy [114]. In addition, nephrotoxicity is unavoidable in cisplatin therapy, and pretreatment with cepharanthine to overexpress metallothionein may reduce the nephrotoxicity caused by the anticancer drug cisplatin [115]. High doses of cepharanthine have a preventive effect in addressing problems such as leukopenia in patients due to anticancer drug therapy [63,64], especially neutropenia [64]. Furthermore, Ohta T et al. [66] demonstrated that 81.3% of the patients were spared leukopenia after the administration of cepharanthine. Leukocyte counts were significantly restored in mice when anticancer drugs were administered concomitantly with cepharanthine compared with administered alone. Accordingly, concomitant administration of anticancer drugs and cepharanthine can improve leukopenia [67]. Furthermore, the oral effect of cepharanthine appears less significant than that of intravenous administration. A dose of 1 mg/kg cepharanthine shows no recovery effect on lymphocytopenia and thrombocytopenia [64]. However, the intravenous administration of 50 mg/d cepharanthine in combination with the anticancer drug CDDP-ACR-CPA (CAP) promotes the recovery of leukopenia and thrombocytopenia caused by the treatment of ovarian cancer [65]. Nemoto K et al. [164] also showed that cepharanthine significantly accelerated the recovery of leukopenia caused by X-ray (5 Gy) irradiation in mice indirectly and had a promotive effect on the recovery of hematopoietic stem cells. In terms of clinical trial results in treating radiotherapy-induced leukopenia, no adverse events related to cepharanthine were reported. Therefore, cepharanthine is safe and effective in treating radiotherapy-induced leukopenia [68]. As an anti-leukopenic drug, cepharanthine also exhibits significant hematopoietic restorative effects [61]. Damage to hematopoietic cells caused by X-ray (3 Gy) radiation can be restored by oral administration of cepharanthine, which also accelerates the recovery of splenic red marrow hematopoiesis [165]. It has been shown that restoring hematopoietic function by cepharanthine is achieved by stimulating bone marrow stromal cells, producing the cytokine IL-6, and acting as a cytokine inducer, stimulating the production of cytokines by various epithelial cells in vivo. Moreover, this stimulates multipotent hematopoietic progenitor cells and their progeny as growth and differentiation factors, thereby restoring hematopoietic function [61], but no effect was found to improve thrombocytopenia [62].

The above results show that cepharanthine has good efficacy in alleviating the side effects of cancer treatment. It can improve the patient’s condition in many ways and has good therapeutic prospects in alleviating the side effects of radiotherapy and chemotherapy.

#### 2.2.6. Enhancement of Immune Activity

Cepharanthine is an immune modulator [53] that can achieve antitumor effects by enhancing sequential immune mechanisms [130]. In an experimental mouse model of bilaterally transplanted solid tumors, genistein played an essential role in sequential immune mechanisms through the induction of Lyt-1-positive cytotoxic cells and T cells. Moreover, this allows tumors both right and left sides to be cured after intratumoural administration to the right side only [166].

Cepharanthine can regulate multiple signaling pathways of abnormally activated T cells, such as NF-κB, caspase cascade, cell cycle, MAPK, and PI3K/Akt/mTOR, in a low-toxicity manner [50]. By affecting these signaling pathways, it can regulate immune activity and pathways related to these signaling pathways, such as apoptosis, cell cycle blockade, autophagy, cellular drug resistance, and multifaceted tumor suppression. Therefore, cepharanthine has broad application prospects as an effective antitumor drug (Table 2).

### 2.3. Anti-Inflammatory to Prevent Shock

Cepharanthine can reduce inflammatory responses both in vitro and in vivo [48], probably by exerting different intracellular and nuclear effects and membrane effects [7]. Moreover, it could inhibit NF-κB and other signaling pathways, reduce proinflammatory cytokine levels, and scavenge free radicals and antioxidants to achieve anti-inflammatory and anti-shock effects (Figure 4).

#### 2.3.1. Reduces the Levels of Proinflammatory Cytokines

Cepharanthine reduces pro-inflammatory cytokine levels by inhibiting NF-κB and other signaling pathways. By interfering with multiple metabolic axes, activating AMP, and inhibiting NF-κB [7], cepharanthine was found to inhibit NF-κB activation by inhibiting the IKK pathway in a rat model, which in turn inhibited lipopolysaccharide-induced increases in serum cytokine levels in rats [167]. Huang H et al. [48] demonstrated the anti-inflammatory effect of cepharanthine, not only in vitro but also in vivo, by inhibiting the NLRP3 signaling pathway, thereby reducing the expression levels of the proinflammatory cytokines IL-1β and TNF-α, and alleviating diabetic nephropathy [46]. In vitro, it not only dose-dependently inhibited lipopolysaccharide LPS-induced NF-κB activation, IκB-α degradation and phosphorylation of ERK, JNK, and p38 but also inhibited the release of lipopolysaccharide LPS-stimulated pro-inflammatory cytokines TNF-α, IL-6 and IL-1β in RAW 264.7 cells. In addition, in vivo lung histopathological analysis showed that cepharanthine attenuated lung histopathological changes and was also able to downregulate the levels of proinflammatory cytokines, including TNF-α, IL-1β, and IL-6. In the absence of T or B cells and under normal conditions, cepharanthine enhanced lipopolysaccharide-induced cytokine production by macrophages and histidine decarboxylase (HDC) activity [168]. It can also inhibit vascular inflammation by suppressing the activation effect of macrophages and inhibiting the production of prostaglandins (PGE2) as well as nitric oxide (NO) [169]. Nader Pazyar et al. [170] also showed that cepharanthine could inhibit the activation of neutrophils and the expression of NF-κB, IL-8, VEGF, and NO. Moreover, cepharanthine also interacted with cytokines such as TNF-a, IL-1b, IL-6, and lipid peroxidation. However, cepharanthine does not reduce neutrophil chemotaxis and phagocytosis [171].

In a mouse lipopolysaccharide-induced mastitis model, cepharanthine was found to reduce neutrophil infiltration, inhibit myeloperoxidase activity, and reduce the levels of the proinflammatory cytokines TNF-α, IL-1β, and IL-6. Furthermore, it also inhibits phosphorylation of the NF-κB p65 subunit and degradation of its inhibitor IκB-α to relieve mastitis in mice [49]. However, Western blot analysis by Aota K et al. [172] showed that cepharanthine did not interfere with the NF-κB pathway, although it was able to block the phosphorylation of JAK2 and STAT1, which are stress inflammatory signaling pathways. Taken together, cepharanthine can exert anti-inflammatory effects by at least partially inhibiting the NF-κB pathway and reducing the levels of pro-inflammatory cytokines. All of these factors make cepharanthine a potential therapeutic agent for inflammatory responses.

#### 2.3.2. Antioxidation Effect

Reactive oxygen species (ROS) are among the most potent mediators of inflammation and play an essential role as primary tissue damage agents [171]. Endogenous free radicals, a product of oxidative metabolism in untreated cells, induce DNA damage. Cepharanthine, an effective ROS scavenger [171], reduces the production of superoxide anion (O^2−^) by neutrophils and macrophages [173] and reduces the level of effective ROS, such as O^2−^, H_2_O_2_, and OH·, which are overproduced, especially in tissues and organs at the site of inflammation [173], achieving anti-inflammatory effects.

Cepharanthine is considered to have antiperoxidant activity due to its membrane-stabilizing effect and its ability to scavenge free radicals. A dose of 30 mg/mL cepharanthine inhibited lipid peroxidation in linoleic acid emulsions by 94.6% [174] and was able to inhibit lipid peroxidation in mitochondria and liposomes via Fe^2+^/ADP [175], with effective antioxidant and free radical scavenging activity [176]. Sato E et al. [177] found that the membrane modulator cepharanthine can inhibit superoxide dismutase produced by chemotactic peptides, FMLP, and neutrophils. It can inhibit the production of ROS by stabilizing the plasma membrane, inhibiting PKC and NADPH oxidase activation, superoxide production and luminol-dependent chemiluminescence, and phosphorylation of cytoplasmic proteins [176].

Not only does cepharanthine scavenge free radicals to prevent lipid peroxidation, but it also protects DNA from endogenous oxidants [10]. Cepharanthine effectively scavenges superoxide anions produced by the hypoxanthine-xanthine oxidase reaction [178] and hydroxyl radicals produced by the Fenton reaction [178]. Furthermore, nitric oxide produced by NOC-7 in the presence of the nitric oxide scavenger C-PTIO [178] or nitric oxide NO induced by lipopolysaccharide [45] inhibits inducible nitric oxide synthase (iNOS) and cyclooxygenase (COX-2) expression, significantly inhibiting lipid peroxidation [169]. Sakaguchi S et al. [179] found that cepharanthine could protect the body from various disorders caused by endotoxin by inhibiting NO production during infectious shock, an effect that may be mediated by enhancing the proliferation of fibroblasts to inhibit endotoxin-induced NO in macrophages, thereby preventing lethality or cytotoxicity. Although the direct scavenging of free radicals achieves anti-lipid peroxidation activity, this activity is pH dependent. In acidic solutions, cepharanthine does not interact with free radicals [180]. Goto M et al. [181] indicated that cepharanthine reduced mortality within 24 h of endotoxic shock in a dose-dependent manner.

Daisuke Sawamura [182] showed that cepharanthine inhibited superoxide anion production by both macrophages and polymorphonuclear leukocytes, and superoxide anion production by PEC (which is mainly dependent on the macrophage component) could be inhibited by 34% with 5 µg/mL cepharanthine and that cepharanthine inhibited O^2−^ production by macrophages. Nevertheless, experiments conducted in the xanthine-xanthine oxidase system showed that it did not inhibit the production of O^2−^. This result suggests that it is not a scavenger of O^2−^. It also inhibited the PMN metabolic response by inhibiting O^2−^ [183].

#### 2.3.3. Treatment of Inflammation and Shock

Cepharanthine can reduce lung injury, liver injury, otitis media, stomatitis, and infectious shock. Some studies have also shown that cepharanthine inhibits NF-κB and the genes it regulates, which is closely related to its anti-inflammatory effects [54].

Cepharanthine inhibited tyrosine phosphatase and the initiation step of neutrophil activation [184], inhibited elastase release from isolated neutrophils stimulated with formylmethyl leucine phenylalanine (fMLP) or fosetyl myristate and elevated intra-neutrophil calcium levels. Moreover, it also inhibited protein kinase C or other downstream signaling pathways in neutrophil activation and thereby attenuated sheep inhalation-induced acute lung injury and infectious shock [185]. Furthermore, it significantly attenuated the proinflammatory cytokine response to HS/RES-induced acute lung injury in rats [186]. This process may be related to the HO^−1^ pathway, an antioxidant factor, which has been shown to reduce acute lung injury and infectious shock following smoke inhalation in sheep [185]. In addition, it also significantly reduces the proinflammatory cytokine response to HS/RES-induced acute lung injury in rats [186]. Cepharanthine also treated lethal shock by reducing the release of enzymes from hepatocytes into the circulation, reducing apoptotic cells and, thus, reducing lipopolysaccharide-induced liver injury [187]. It can also treat NLRP3 inflammasomes and reduce inflammatory kidney damage in DN in diabetic nephropathy [188]. It can also inhibit microglial activation and NLRP3 pathway-induced inflammation as well as inhibit 12/15-LOX signaling and reduce oxidative stress to reduce cerebral ischemia/reperfusion (I/R) injury [188]. Following intraperitoneal injection of cepharanthine in a hairy mouse model, cepharanthine was diffusely distributed throughout the middle ear mucosa to transfer to the middle ear and prevent and repair experimental otitis media [189]. In clinical practice, oral administration of 20 mg of cepharanthine daily for 4 weeks or more resulted in 83.3% improvement in stomatitis, 87.0% improvement in oral lichen planus, 77.8% improvement in tongue pain, and 80.0% improvement in leukoplakia [173]. Cepharanthine may also reduce muscle and kidney damage due to limb I/R through anti-inflammatory or antioxidant effects [190]. In conclusion, cepharanthine has excellent anti-inflammatory and anti-injury effects.

### 2.4. Immunomodulatory Effects

The immunomodulatory activity of cepharanthine can be utilized in the fight against tumors and other areas such as vasodilation and autoimmune diseases.

#### 2.4.1. Vasodilatory Effect

The vasodilatory effect of genistein improves circulation. Cepharanthine can improve cutaneous microcirculation in rabbits through vasodilatory effects [191], causing significant and transient vasodilation with vasodilatory (approximately 1 h or longer) effects in the subcutaneous tissue within the rabbit hyaline ear [191]. The immune response-enhancing, peripheral circulation-improving, vasodilatory-modulating activity of cepharanthine suggests that it may be beneficial in the treatment of patients with sickle cells [192].

#### 2.4.2. Treating Allergic Reactions

Regulates multiple signaling pathways of abnormally activated T cells;

Cepharanthine can modulate several signaling pathways that abnormally activate T cells in a low-toxicity manner, such as NF-κB, caspase cascade, cell cycle, MAPK, and PI3K/Akt/mTOR. It is also therapeutically important in autoimmune diseases or refractory T cell acute lymphoblastic leukemia (T-ALL) with functional p-glycoprotein disease [50]. Cepharanthine can block DRβ1-Arg74 receptors and can block T-cell activation through thyroglobulin peptide Tg2098, among others [52], and, as a DRβ1-Arg74 receptor blocker, it is also able to block the expression of TSH receptor peptide through TSHR presentation and T-cell response, which is essential for the treatment of Graves’ disease [51]. Uto T et al. [53] showed that 1–5 μg/mL cepharanthine could inhibit antigen uptake by dendritic cells and reduce the production of interleukin-6 and tumor necrosis factor-α in lipopolysaccharide-stimulated dendritic cells. This suggests that cepharanthine has excellent therapeutic potential for autoimmune diseases and allergies.

Treating allergic reactions;

Cepharanthine can treat allergic reactions not only by modulating cytokines but also by inhibiting the release of histamine from mast cells in sensitized animals [193], altering membrane stability [193] and stimulating pituitary gland pro-adrenal function [194] to achieve anti-allergic effects.

Cepharanthine is clinically effective in allergic rhinitis and is a potential clinical agent for patients with nasal allergies [195,196]. The ability to affect the inhibition of HDC activity in mast cell-sufficient mice by a single administration of cepharanthine [3], which acts as an inhibitor of histamine release from mast cells, was investigated by studying the effect of cepharanthine on gastric motor excitatory responses in dogs. It was found that cepharanthine was able to inhibit histamine release from histamine-secreting cells in the gastric mucosa [197]. However, Nakamura K et al. [198] showed that cepharanthine did not inhibit histamine release at the cellular level.

In a rat model of experimental allergic rhinitis, it was found that 0.1 mg/mL cepharanthine inhibited the increase in lysosomal enzyme activity, demonstrating that cepharanthine achieves membrane stability by reducing membrane elasticity, which in turn exerts an anti-allergic effect [193]. In dendritic cells treated with cepharanthine, the mitochondrial membrane potential was reduced, cysteine levels were increased, and DNA was broken. Thus, all of these findings indicate that cepharanthine can induce apoptosis and may be a potential drug for the treatment of dendritic cell-mediated allergic diseases [199]. Cepharanthine can also promote corticosterone secretion through stimulation of pituitary pro-adrenal function to achieve anti-allergic effects [194].

#### 2.4.3. Platelet-Related Diseases

Chronic immune thrombocytopenia;

Several studies have shown that cepharanthine can inhibit thrombocytopenia and treat idiopathic thrombocytopenic purpura [200]. In the clinical setting, 40 mg/d cepharanthine by oral administration was able to treat idiopathic thrombocytopenic purpura (ITP) [201], while large amounts of cepharanthine in combination with prednisolone resulted in increased platelet counts [202], thereby reducing the dose of prednisolone used [203]. Cepharanthine also acts in combination with danazol, ascorbic acid, cimetidine, [204] and corticosteroids (CSs) [205] in the treatment of idiopathic thrombocytopenic purpura. In addition, cepharanthine improves the platelet production process [206] and treats periodic thrombocytopenia [207]. Cepharanthine has been used to treat progressive thrombocytopenia due to abnormalities in the immune system. It has been successfully used to treat an elderly Japanese patient with multiple myeloma and immune thrombocytopenia [208].

Inhibition of platelet aggregation, and platelet activation;

Cepharanthine was able to inhibit platelet activation [209] and collagen-induced platelet aggregation [145] in a dose-dependent manner by inhibiting phospholipase A2 activation [4] but not platelet aggregation induced by other stimuli, such as thrombin and arachidonic acid [145]. Akiba S et al. [210] further showed that cepharanthine vincristine could inhibit the release of arachidonic acid by uncoupling GTP-binding proteins from the enzyme and thereby inhibiting receptor-mediated activation of phospholipase A2. Calcium ions play an essential role in platelet aggregation, and cepharanthine can inhibit the function of calcium channels. It can also alter the sensitivity of phospholipids to phospholipase A2 enzymatic cleavage. Moreover, it dose-dependently inhibits calcium inward flow and collagen-induced platelet aggregation in rabbits [211]. It also inhibits changes in the membrane state by inhibiting physicochemical reactions such as accelerated oxygen consumption, the release of membrane-bound Ca^2+^, the release of Ca^2+^ to the extracellular medium and the reduction in membrane potential, thereby affecting platelet activation and aggregation [212].

#### 2.4.4. Other Autoimmune Diseases

Cepharanthine is effective in treating a range of autoimmune diseases, such as autoimmune thyroid disease, toxic diffuse goiter (Graves’ disease), Sjögren’s syndrome, and alopecia areata. Cepharanthine mainly treats autoimmune thyroiditis-related disorders [22,52,213], such as hypothyroidism, by blocking T-cell activation [213]. Cepharanthine is used to treat autoimmune thyroid disease by blocking T-cell activation through the thyroglobulin peptide Tg2098, a peptide that binds to the arginine-containing HLA-DR variant (DRβ1-Arg74) that causes autoimmune thyroid disease (AITD) [52]. Li CW et al. [51] also showed that cepharanthine could block TSHR presentation and T-cell responses and block the expression of TSH receptor peptides via HLA-DR3 for the treatment of Graves’ disease. In autoimmune polyglandular syndrome type 3 variant (APS3v), which occurs concurrently with type I diabetes (T1D) and autoimmune thyroiditis (AITD), cepharanthine may also have therapeutic potential in such patients. It can block T-cell activation by thyroid/islet peptides under both in vivo and in vitro conditions [22]. Azuma M et al. [214] showed that cepharanthine can also prevent structural destruction of salivary gland vesicles in patients with Sjögren’s syndrome by inhibiting NF-κB. For the possible autoimmune pathogenesis of alopecia areata, Morita K et al. [69] found that cepharanthine could act in combination with topical dibutyl squarate (SADBE) to treat alopecia areata.

### 2.5. Inhibition of Bone Resorption

Cepharanthine can inhibit NF-κB signal pathway to further restrain the formation of osteoclasts induced by its receptor activating factor ligand (RANKL), as well as inhibit the osteoclast differentiation marker genes [40], c-Jun N-terminal kinase (JNK), and phosphatidylinositol 3-kinase (PI3K)-AKT signaling pathways to inhibit bone resorption in vivo, as it is not caused by enhancing bone formation. Thus, cepharanthine protects against bone loss due to estrogen deficiency and may be a potential drug for the treatment of osteoporosis [39]. Cepharanthine inhibits not only bone resorption but also osteolysis. It prevents periprosthetic osteolysis by inhibiting osteoclast production and reducing the ratio of NF-κB receptor activator ligand (RANKL)/osteoprotegerin (OPG) caused by wear particles [41].

### 2.6. Cell Proliferation—Treatment of Hair Loss

Cepharanthine has the potential to treat hair loss. The proliferative activity of hair cells was increased according to the treatment of cultured mouse skin hair cells with 0.01–0.1 μM cepharanthine [14]. Furthermore, the common use of cepharanthine and minoxidil promoted hair cell proliferation, differentiation, and keratinization [15]. One study demonstrated that cepharanthine could increase the concentration of Ca^2+^ in human hair papilla cells (HDPC cells), increasing the expression of HIF-1α and HIF-2α in HDPC and inducing the expression of HIF-responsive genes. Moreover, it promotes the proliferation of HDPC and increases the expression of its vascular endothelial growth factor (VEGF) to restore hair growth [71]. External application of cepharanthine can also promote the production of IGF-I, inhibiting nonscarring alopecia and other alopecia diseases [70].

### 2.7. Treatment of Poisonous Snake Bites

Cepharanthine is effective in the treatment of poisonous snakebites. In a mouse model, the lethal effect of viper venom injected with a lethal dose of four to five times was significantly inhibited by the injection of cepharanthine. However, the oral administration of cepharanthine did not have a significant effect. Clinically, it was also found that the injection of cepharanthine within 6 h after a viper bite reduced the average number of days of treatment. It also inhibits capillary hemorrhage and improves circulatory system activation to suppress snake venom [57]. However, snake venom treatment with cepharanthine is less effective than anti-venom treatment, and the length of hospital stay is significantly shorter in patients treated with anti-venom therapy than in those treated with cepharanthine [56]. Therefore, cepharanthine can be used with anti-venom and methylprednisolone [59] to treat snakebites.

### 2.8. Other Roles

#### 2.8.1. Non-Covalent Interaction with Telomeric RNA G-Quadruplexes

As telomeric RNA is associated with telomerase activity, the search for ligands to regulate the structure of the RNA G-quadruplexes may be necessary for the regulation of telomerase activity. The RNA G-quadruplexes structure of telomeres has a high binding affinity with cepharanthine, and it is also associated with telomeric RNA and telomerase activity. Consequently, this role may contribute to the regulation of telomeric RNA and telomerase activity [215].

#### 2.8.2. Anti-Atherosclerotic

Atherosclerosis has an essential relationship with the inflammatory response, which produces a range of cytokines and chemokines that can drive the development of atherosclerosis. In addition to treating tumors and inflammation, macrophage modulation by cepharanthine can also have an anti-atherosclerotic effect. The production of large amounts of nitric oxide (NO) by macrophage activation-induced nitric oxide synthase and the proliferation and migration of vascular smooth muscle cells in response to mitogens contribute to atherosclerosis. Furthermore, inhibiting cytokine production and blocking the proliferation and migration of vascular smooth muscle cells (VSMCs) after macrophage activation is an important therapeutic strategy for preventing atherosclerosis [169]. By inhibiting platelet-derived growth factor (PDGF-BB) production and NF-κB translocation, cepharanthine can achieve a dose-dependent inhibition of the proliferation and migration of VSMCs induced by PDGF-BB. This makes cepharanthine a potentially effective agent for the prevention and treatment of atherosclerosis [169].

#### 2.8.3. Inhibition of Intimal Hyperplasia

In a dog model with a superior vena cava replacement using a Teflon fluorocarbon resin grafts, it was found that treatment with cepharanthine resulted in a lower incidence of luminal obstruction and reduced intimal hyperplasia compared with the control group [216]. Additional studies in 80 adult mongrel dogs with Teflon grafts implanted into the superior vena cava also revealed a significant reduction in graft obstruction following treatment with cepharanthine. Moreover, it showed good long-term outcomes in 18 patients who underwent various venous reconstructions with subsequent reconstruction of the superior vena cava prosthesis. Consequently, all these findings suggest that cepharanthine may prevent endothelial hyperplasia [217].

#### 2.8.4. Ion Channel Inhibitor

Cepharanthine can regulate the efflux pump and has an inhibitory effect on the sodium-potassium pump. Cepharanthine has a weak inhibition of the activity of adenosine triphosphatase (Na^+^, K^+^-ATPase) [218] and is also able to inhibit the loss of erythrocyte K^+^ caused by toxic substances of adenosine triphosphatase activated by the sodium-potassium pump through a protective effect on the sodium-potassium pump [219]. It was also shown that the effect of cepharanthine on adenosine triphosphatase occurs through the inhibition of lipid peroxidation induced by reactive oxygen species to prevent the inactivation of ATPase caused by peroxidation. In the absence of Fe^3+^, cepharanthine can act as an inhibitor of adenosine triphosphatase by antagonizing ascorbic acid [219].

The effect of cepharanthine on inhibiting calcium channels is through the physical alteration of lipid properties to inhibit calcium inward flow [211], and it is believed to be the reason why cepharanthine can inhibit a variety of virus infection. Moreover, it also inhibits a range of calcium-induced responses in a dose-dependent manner to inhibit membrane permeability transition (MPT) [220].

#### 2.8.5. Inhibition of Neurodegenerative Diseases (NDDs)

Cepharanthine can inhibit neurodegenerative diseases (NDDs) by inhibiting lipopolysaccharide (LPS). Activation of microglial cells in the brain has been considered to be associated with various NDD. Cepharanthine inhibits LPS-induced microglial cell activation associated with various NDDs and the release of cytokines (TNF-α, IL-1β, and IL-6) from microglia to achieve therapeutic effects in NDDs associated with microglial cell activation [221].

#### 2.8.6. Sickle-Cell Anemia

Cepharanthine has an anti-sickle anemia effect in vitro, and it inhibits 50% of irreversible sickle cell formation at a dose of 15 µM, a concentration well below that required to inhibit sickle disease in vitro [222]. It can also dose-dependently alter the shape of human red blood cells [223]. Moreover, cepharanthine did not have a neuroleptic effect compared with the anti-neuroleptic drug chlorpromazine. Moreover, its effects of improving circulation, enhancing immune activity and vasodilation, and inhibiting platelet aggregation have a positive effect on patients with sickle-cell anemia [192].

#### 2.8.7. Treatment of Amyloidosis and Alzheimer’s Disease

Cepharanthine is therapeutically effective in primary limited cutaneous amyloidosis [224]. It has been shown that biphasic amyloidotic symmetrical papular lesions (amyloid moss) were significantly flattened after 6 months of topical treatment with cepharanthine [225]. Cepharanthine was able to selectively inhibit the binding of β-amyloid oligomers to EphB2 [226] and improve the deposition of β-amyloid (Aβ) and NLRP3 [227], which may be a potential treatment for Alzheimer’s disease. In a rat model of Alzheimer’s disease, cepharanthine was also found to improve β-amyloid (Aβ) deposition. Moreover, its joint use with dexmedetomidine (DEM) improved symptoms in neurological function scores and cognitive function in Alzheimer’s rats [227].

## 3. Safety and Bioavailability of Cepharanthine

### 3.1. Bioavailability

Cepharanthine has been used for many years in Japan to treat acute chronic diseases [54,122] and is clinically utilized in both oral and injectable administration forms such as tablets and powders for oral administration [5]. After being absorbed, cepharanthine is mainly distributed throughout the liver, kidneys, spleen, and lungs [5].

The absorption of cepharanthine in the human intestine was investigated using a monolayer Caco-2 (human colon adenocarcinoma cell line) cell model of the intestinal epithelium. It was found that cepharanthine could be completely absorbed by the intestinal epithelium [228].

In the liver, cepharanthine is extensively metabolized, and the absolute bioavailability of cepharanthine after oral administration is 6–9% [229]. Following a single intravenous dose of 50 mg, blood concentrations of cepharanthine in human plasma decreased rapidly within 2 h, with a mean maximum concentration of 135.9 ± 66.9 ng/mL occurring at 0.75 ± 0.21 h after dosing [229]. The peak blood concentrations were 153.17 ± 16.18 ng/mL and 46.89 ± 5.25 ng/mL, and the t1/2 values were 6.76 ± 1.21 h and 11.02 ± 1.32 h with injectable and oral administration, respectively. However, oral administration of cepharanthine reached its peak at approximately 2.67 h, and the absolute bioavailability was approximately 5.65 ± 0.35%, which indicated that oral administration of cepharanthine resulted in poor absorption and that the distribution and elimination of gentian vine were slow in rats.

The poor solubility and low bioavailability of cepharanthine seriously affect its efficacy. In order to improve the clinical efficacy of cepharanthine, a variety of methods and dosage forms have been designed and used [230,231,232]. Gao P et al. [230] designed a self-emulsifying drug delivery system (SEDDS) loaded with cepharanthine for administration, and the bioavailability of CEP-SEDDS in mice was 203.64% compared with cepharanthine. Gao et al. [231] enhanced the accumulation of cepharanthine nanoparticles in the lungs by wrapping cepharanthine nanoparticles through macrophage membranes. Dou et al. improved the solubility of cepharanthine by acidic carriers and increased the bioavailability of cepharanthine 68-fold by pulmonary delivery compared with oral delivery [232].

### 3.2. Safety

As mentioned previously, cepharanthine inhibits cell proliferation by inhibiting the NF-κB signaling pathway. On the one hand, this is one of the important mechanisms for its antitumor effect and inhibition of abnormal cell proliferation induced by viral infection. On the other hand, the inhibition of NF-κB activity may cause cepharanthine to show greater drug toxicity on some human cells. For example, the CC50 of cepharanthine on MRC-5, Huh7, and A549-ACE2 cells was as low as at 10.54 µM [77], 24 µM [20], and 30.92 µM [20], respectively. However, the effective concentration is usually far less than the maximum safe concentration.

Cepharanthine has been used in Japan since 1950 [7], and no serious side effects were found with cepharanthine in current clinical use [5,192]. Sato T et al. mentioned two patients who received high doses of cepharanthine orally (6 g orally per day for 23 consecutive days and 3 g orally per day for 35 consecutive days) and one patient who received 40-60 mg of cepharanthine intravenously per day for 2 months, and none of these patients reported any side effects [192]. Y Arai et al. used cepharanthine in combination with intraarterial injection of vinblastine and adriamycin (or epirubicin) to treat 6 patients with metastatic renal cell carcinoma without significant side effects [233]. Yukunori Korogi et al. analyzed the therapeutic effects of two administration methods of cepharanthine (intravenous, oral) on patients with prostate cancer, and no obvious adverse reactions were observed in the two administration methods [114]. Yoshiki Miyachi et al. combined with oral cepharanthine (3–30 mg/day) and topical SADBE (squaric acid dibutylester) to treat severe alopecia areata, and no side effects of cepharanthine were reported in this study [69]. Additionally, there is no cepharanthine related side effects was reported in two clinical trials (study of oral high/low-dose cepharanthine compared with placebo in non-hospitalized adults with COVID-19, NCT05398705; Cepharanthine trials, JPRN-jRCTs061180072) (https://trialsearch.who.int/ (accessed on 14 June 2023)). However, commercial cepharanthine tablets have been described as having side effects, occasionally causing mild gastrointestinal upset [234].

## 4. Conclusions and Prospectives

In the treatment of cancers, cepharanthine induces apoptosis [6,13,17,44,47,60,117,118,122,235,236,237,238,239], inhibits autophagy [8,42,124], causes cell cycle arrest [9,17,82,119], and inhibits angiogenesis [80,102,130,240], which in turn inhibits tumor cell proliferation [107]. Moreover, it enhances the activity of anticancer drugs by disrupting plasma membrane function [147,241,242,243], increases tumor cell sensitivity [8,43,118,135,152,153,154,155,160,244,245,246,247,248,249,250], and reverses multidrug resistance to Adriamycin [9,43,140,150,243,244,251,252], erythromycin [150], vincristine [150], and paclitaxel [151,253]. Furthermore, it also inhibits tumor metastasis by inhibiting cell migration [13,117,126] and is able to treat radiation-induced injuries by reducing damage from chemoradiotherapy [114,115,254], leukopenia [61,62,63,64,65,66,67,68,161,164], thrombocytopenia [65,255], radiation-induced hematopoietic damage [165], and dry mouth [162,163].

In the anti-inflammatory field [7,44,45,46,47,48,49], cepharanthine is able to treat mastitis [49]; otitis media [189]; injuries in the lung [185,186,256], liver, and kidney [187,256,257]; diabetic nephropathy [188]; and renal injury [258]. It inhibits NF-κB activation; IκB-α degradation; ERK, p38, and JNK phosphorylation; and the downregulation of proinflammatory cytokine levels, thereby exerting anti-inflammatory effects [48]. Anti-inflammatory effects can also be achieved by reducing the overproduction of potent reactive oxygen species in tissues and organs, especially at sites of inflammation [171].

In immunomodulation, it can achieve immunomodulatory effects by modulating multiple signaling pathways of abnormally activated T cells [50]. In the treatment of chronic immune thrombocytopenia [200,202,205,206,207,208,259], platelet aggregation [192,260], platelet activation [4,209], autoimmune thyroid disease [22,52], toxic diffuse goiter (Graves’ disease) [51], and other autoimmune diseases such as arthritis [40], Sjögren’s syndrome [214], and pemphigus vulgaris [69] have been studied.

Regarding resistance to pathogens, cepharanthine has been used against parasites such as *Plasmodium* [28,30,31,32,33,34] and *Trypanosoma cruzi* [29,103], bacteria such as methicillin-resistant and gentamicin-resistant *Staphylococcus aureus* [36], *Mycobacterium tuberculosis* [37], *Mycobacterium leprae* [38], and coronaviruses such as SARS-CoV-2 [20,21,24,27,72,73,84,85,86,87,88,89,90]. Moreover, it has also been used for human coronaviruses SARS-CoV [73,92], MERS-CoV [92], HCoV-OC43 [77], SARS-CoV-2-like GX_P2V [21,93], porcine coronaviruses SADS-CoV [95] and PEDV [94], and other viruses, including HIV [26,74,75,76,96], HSV-1 [82,97], EboV [20], Zika [20], PRRSV [78], PCV2 [98], HYLV-1 [79], and CV-B3 [99], all of which have some degree of inhibition and, therefore, may be potential broad-spectrum antiviral agents.

Among other fields, cepharanthine can inhibit bone resorption and act as a potential anti-osteoporosis agent [39]. It may also potentially be used for the treatment of late-onset neurodegenerative neuromuscular diseases by inhibiting the release of cytokines [221], which can bind to G-quadruplex nucleic acid structures, providing insight into the regulation of telomeric RNA and telomerase activity [215]. Moreover, it can also inhibit endothelial hyperplasia [216,217], treat venomous snake bites [55,56,57,58,59], treat amyloidosis [224,225,227,261], exert potential anti-atherosclerotic effects [169], ameliorate sickle-cell anemia [192,222,223], and be a potential drug for Alzheimer’s disease [98,227].

In addition to its many clinical activities, preclinical studies have also demonstrated a wide range of potent activities, suggesting that cepharanthine has the potential to treat other illnesses as well as a variety of diseases, such as viral, bacterial and parasitic infections, autoimmune diseases, Alzheimer’s disease, osteoporosis, inflammation, and shock. Moreover, it may even have a role in prolonging life and slowing down the process of aging. Therefore, cepharanthine has broad application prospects and still has endless potential as a medicine with a long history (Table 3).

The clinical application of cepharanthine is limited due to its poor water solubility and low oral bioavailability [262]. At present, the commercial dosage forms of cepharanthine are mainly ordinary tablets. Liang et al. [263] proposed a variety of formulations that can improve the bioavailability of cepharanthine. These include oral formulation (oral disintegrating tablets, dropping pills), injections, and pulmonary drug delivery systems—DPIs, nano-formulations (liposomes and nanoparticles). However, there are still some defects in these delivery methods. Oral disintegrating tablets are not easy to store. Dropping pills lack quality standards for administration and cannot be approved for marketing at this time. The solubility of cepharanthine in injections is low and its use is limited. Pulmonary drug delivery is limited by the solubility of cepharanthine and requires attention to formulation design. Nano-formulations, while improving solubility and targeting, may stimulate the body’s immune system. Although cepharanthine has pharmacological activities such as anti-pathogen, anti-tumor, and anti-inflammatory, its wide application is limited by its low bioavailability. Therefore, follow-up studies should be mainly aiming at the optimization of the administration mode and dosage form of cepharanthine, which plays a decisive role in the revival of the old drug “cepharanthine”.

## Figures and Tables

**Figure 1 molecules-28-05019-f001:**
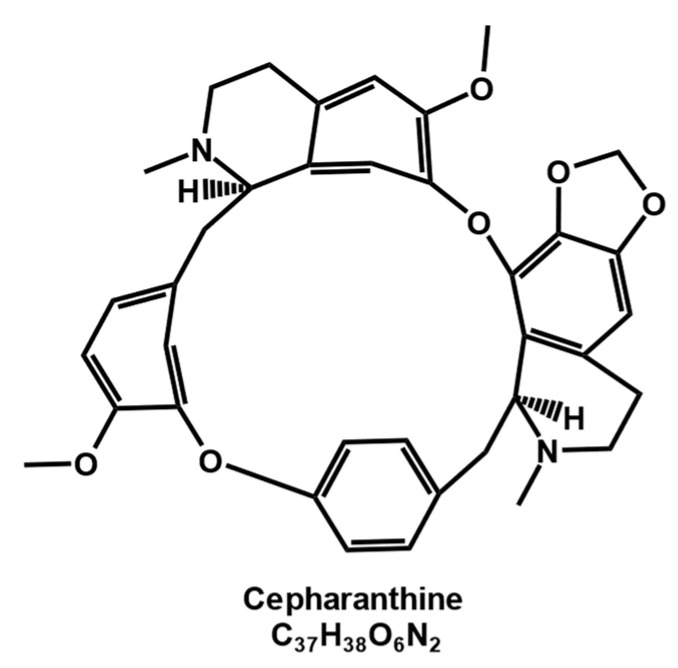
Chemical structure of cepharanthine.

**Figure 2 molecules-28-05019-f002:**
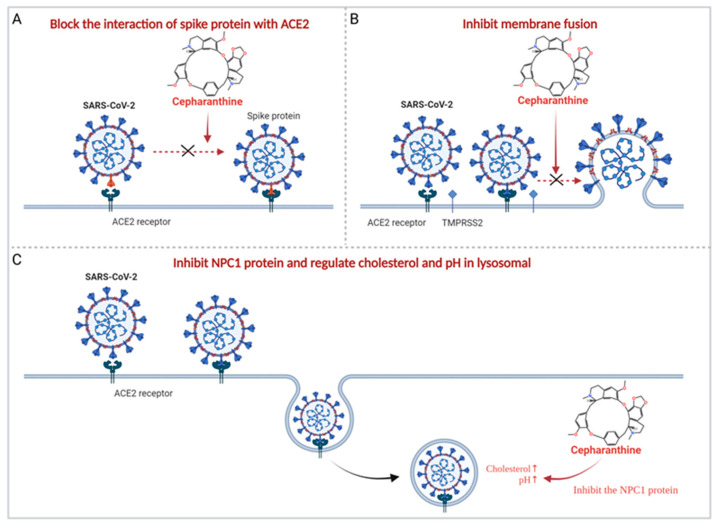
Main potential mechanisms of cepharanthine against SARS-CoV-2. (**A**) Cepharanthine blocks the interaction of SARS-CoV-2 spike protein with ACE2, thereby inhibiting the entry of SARS-CoV-2. (**B**) Cepharanthine inhibits SARS-CoV-2 spike protein-mediated membrane fusion and plays an antiviral role. (**C**) Cepharanthine inhibits NPC1 protein, regulates lysosomal cholesterol and pH, and disrupts cellular/lysosome lipid homeostasis, thereby exerting the activity against SARS-CoV-2.

**Figure 3 molecules-28-05019-f003:**
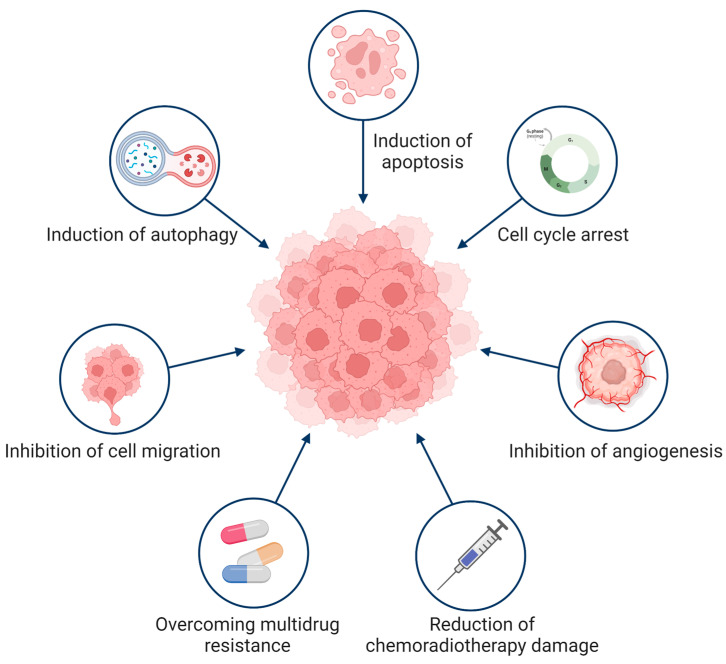
Antitumor mechanism of cepharanthine. Cepharanthine exerts antitumor effects through a variety of ways.

**Figure 4 molecules-28-05019-f004:**
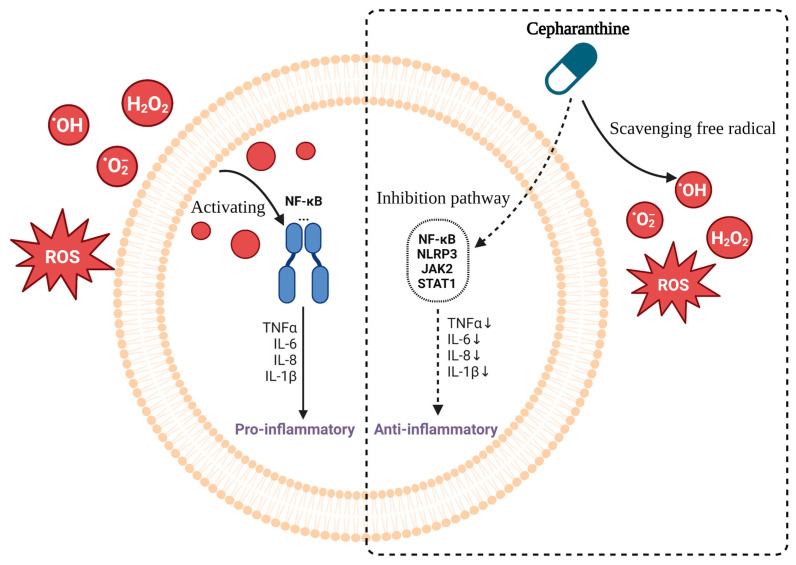
Mechanism of cepharanthine in inhibiting inflammatory response. Free radicals activate NF-κB pathway to produce inflammatory response (Left). Cepharanthine reduces inflammation responses by scavenging free radical inhibitory NF-κB and other signaling pathways, thereby reducing the level of pro-inflammatory factors (Right).

**Table 1 molecules-28-05019-t001:** Antiviral activity of cepharanthine.

Virus	Cell	Antiviral Effect
SARS-CoV-2 [100]	A549	EC50 = 0.15 µM
SARS-CoV-2 [20]	hACE2 mice	10 mg/kg cepharanthine effectively improved lung injury and inflammation.
GX_P2V [93]	Vero E6	EC50 = 0.98 µM; CC50 = 39.30 µM; SI = 39.91
SARS-CoV-2 S pseudovirus (G614) [101]	293T-ACE2	EC50 = 0.351 µM
	Calu3	EC50 = 0.759 µM
	A549-ACE2	EC50 = 0.911 µM
SARS-CoV-2 S pseudovirus (D614) [101] SARS-CoV-2 S pseudovirus (N501Y.V1) [101] SARS-CoV-2 S pseudovirus (N501Y.V2) [101]	293T-ACE2	EC50 = 0.0537 µM
	293T-ACE2	EC50 = 0.047 µM
	293T-ACE2	EC50 = 0.140 µM
SARS-CoV S pseudovirus [101]	293T	EC50 = 0.0417 µM
MERS-CoV S pseudovirus [101]	293T	EC50 = 0.140 µM
SARS-CoV [99]	Vero E6	EC50 = 6.0 μg/mL
HCoV-OC43 [77]	MRC-5	EC50 = 0.83µM
PEDV [77]	Vero	EC50 = 2.53 µM
	Piglet	11.1 mg/kg cepharanthine effectively reduced the PEDV load, attenuated histopathological changes, and reduced PEDV damage in piglets.
SADS-CoV [95]	Huh7	EC50 = 0.79 µM
HIV-1 [75]	/	EC50 = 0.026 µM
EBOV Δ VP30-GFP [20]	/	EC50 = 0.42 μM
ZIKV(MR766) [20]	/	EC50 = 2.19 μM
HSV-1 [25]	/	EC50 = 0.835 μg/mL
PRRSV [78]	Marc-145	10 μM cepharanthine reduced the TCID50 of PRRSV by 5.6 times.
PCV2 [98]	PK-15	Cepharanthine inhibited PCV2 infection in a dose-dependent manner, and 0.00075 mg/mL cepharanthine significantly reduced the virus expression.
HTLV-1 [79]	/	Synergistic inhibition of HTLV-1 in combination cepharanthine with TMNAA.
CV-B3 [99]	/	1.25–6.25 μg/mL cepharanthine had a high CVB3 inhibitory activity.

The inhibitory activity of cepharanthine against different viruses and experimental subjects were summarized.

**Table 2 molecules-28-05019-t002:** Summary of the antitumor activity of cepharanthine.

Cell Line	Test Concentration	Mechanism of Action
KHM-11 and 12PE cells [17]	10 μM	Induce apoptosis by regulating ROS, Bax and caspase-3; Block the cell cycle by inducing CDK inhibitors and down-regulating CDK.
Primary effusion lymphoma (PEL) derived cell lines (BCBL-, TY-1, and RM-P1) [12]	10 μg/mL	Inhibit the activation of NF-κB; Induce the apoptosis of PEL cell line.
Hep3B and HCCLM3 cells [13]	20 μM	Induce apoptosis through the activation of caspase-9/3; Regulate amino acid metabolism.
T98G and U87MG cells [123]	15 µg/mL	/
U251MG cells [123]	15 µg/mL	Induce apoptosis via the caspase cascade.
MDA-MB-231 and MCF-7 cells [42]	4 μM	Impair autophagosome–lysosome fusion by mediation the downregulation of MYO1C.
MCF-7 cells [81]	10 μM	Induce autophagy and apoptosis by inhibiting the AKT/mTOR signaling pathway.
MDA-MB-231 cells [81]	7 μM	Induce autophagy and apoptosis by inhibiting the AKT/mTOR signaling pathway.
Hela cells [124]	10 μM	Enhance autophagic flux and autophagosome formation via the AMPK-TSC2-mTOR signaling pathway.
NCI-H1975 cells [8]	20 μM	Block autophagosome–lysosome fusion; Inhibit lysosomal cathepsin B and cathepsin D maturation.
HSC2, HSC3, and HSC4 cells [118]	10 μg/mL	Promote the mitotic death by radiation; Inhibit DNA double-strand break (DSB) repair after radiation.
HSC3 cells [118]	5 μg/mL	Promote the mitotic death by radiation; Inhibit DNA double-strand break (DSB) repair after radiation.
Human adenosquamous cell carcinoma cell line (TYS) [119]	10 μg/mL	Induce G1 arrest via expression of p21Waf1 and apoptosis through caspase 3.
Human osteosarcoma cell line SaOS2 [9]	3.18 μM	Inhibit the STAT3 signaling pathway.
CaOV-3 cells [19]	10 μM	Increase the expression of p21Waf1 protein; Decrease the expression of cyclins A and D proteins and trigger apoptotic cell death.
OVCAR-3 cells [19]	20 μM	Increase the expression of p21Waf1 protein; Decrease the expression of cyclins A and D proteins and trigger apoptotic cell death.
Human umbilical vein endothelial cells (HUVECs) and Human dermal microvascular endothelial cells (HMVECs) [130]	10 μg/mL	Enhance a sequential immune mechanism; Inhibit angiogenesis in tumors.
B88 and HSC3 cells [80]	2 μg/mL	Inhibit the expression of VEGF and IL-8 involved in the blockade of NF-κB activity.
K562 cells [43]	10 µM	Reverse P-gp mediate MDR; Inhibit the acidification of organelles.
SaOS2-AR cells [140]	5.5 μg/mL	Inhibit adriamycin (ADR) resistance on ADR-induced apoptosis and necrosis.
NIH 3T3 cells [142]	1 μg/mL	Improve the drug sensitivity of tumors resistant to adriamycin (ADR).
P388 leukemia (P388/R) cells [143]	3.5 μg/mL	Enhance the antitumor activity of doxorubicin (DOX).
A549 and GR cells [148]	3 μg/mL	Increase the sensitivity to gemcitabine in A549/GR cells by inhibiting the MRP7 expression.
KKU-M213 and KKU-M214 cells [113]	10 μg/mL	Inhibit the metastatic migration and invasion of human CCA cell lines; Inhibit the activation of NF-κB.

The above table summarizes the action dose and mechanism of cepharanthine on different cancer cell lines. “/”: no information was found.

**Table 3 molecules-28-05019-t003:** Pharmacological activities of cepharanthine.

Pharmacological Activity	Speculated Application	Potential Mechanisms
Antipathogenic activity	Antiviral therapy: SARS-CoV-2 [20,21,24,27,72,73,84,85,86,87,88,89,90], SARS-CoV [73,92], MERS-CoV [92], HCoV-OC43 [77], GX_P2V [21,93], PEDV [77], SADS-CoV [77], HIV-1 [26,74,75,76,96], HSV-1 [82,97], Ebola virus [20], Zika virus [20], PRRSV [78], PCV2 [78], HTLV-1 [78], CV-B3 [78]; Anti parasitic therapy: *Plasmodium falciparum* [28,31,33], *Trypanosoma cruzi* [103]; Antibacterial therapy: Methicillin- and gentamicin-resistant *Staphylococcus aureus* [36], *Mycobacterium leprae* [36].	Inhibit the fusion of viral with the cell membrane [101], Stabilize plasma membrane fluidity [74], Inhibit the NPC1 protein [84,102], Reverse dysregulated endoplasmic reticulum stress/unfolded protein response and heat shock response [21], Inhibitory activities on NF-κB [74,75,76], Target the STING/TBK1/P62, the PI3K/Akt and p38 MAPK signaling pathways [25,82].
Antitumor activity	Treatment of Ehrlich’s ascites tumor [106], Primary exudative lymphoma [12], Hepatocellular carcinoma [13], Breast cancer [81], Human oral squamous carcinoma [80,118], Human adenosquamous cell carcinoma [119], Human osteosarcoma [9], Non-small cell lung cancer [8], Lewis lung cancer [157], Bilaterally transplanted solid tumors [166].	Induce apoptosis [6,13,17,44,47,60,117,118,122,235,236,237,238,239], Inhibit autophagy [8,42,124], Cause cell cycle arrest [9,17,82,119], Inhibit angiogenesis [80,102,130,240], Disrupt plasma membrane function [147,241,242,243], Increase tumor cell sensitivity [8,43,118,135,152,153,154,155,160,244,245,246,247,248,249,250], Reverse multidrug resistance [9,43,140,150,243,244,251,252], Reduce damage from chemoradiotherapy [114,115,254].
Anti-inflammatory	Treatment of Mastitis [49], Otitis media [189], Injuries in the lung [185,186,256], Injuries in the liver and kidney [187,256,257], Diabetic nephropathy [188], Renal injury [258].	Inhibit NF-κB activation, IκB-α degradation, ERK, p38, and JNK phosphorylation [48], Reduce proinflammatory cytokine levels [48], Reduce the overproduction of oxygen species [171].
Immunomodulation	Treatment of Chronic immune thrombocytopenia [200,202,205,206,207,208,259], Platelet aggregation [192,260] and Platelet activation [4,209], Autoimmune thyroid disease [22,52], Toxic diffuse goiter (Graves’ disease) [51], Other autoimmune diseases: Arthritis [40], Sjögren’s syndrome [214] Pemphigus vulgaris [69].	Modulate signaling pathways of abnormally activated T cells [50], Inhibit NF-κB signaling pathway [7,167], Reduce proinflammatory cytokine levels [49,171], Scavenge free radicals and antioxidants [171,173].
Others	A potential anti-osteoporosis agent [39]; Treatment of late-onset neurodegenerative neuromuscular diseases [221], Venomous snake bites [55,56,57,58,59], Amyloidosis [224,225,227,261], Ameliorate sickle-cell anemia [192,222,223], Alzheimer’s disease [98,227].	Inhibit bone resorption [39], Inhibit the release of cytokines [221], Inhibit endothelial hyperplasia [216,217], Bind to G-quadruplex nucleic acid structures [215].

The pharmacological activities of cepharanthine mentioned in the review were summarized, and the potential mechanism of action was briefly expounded.

## Data Availability

Not applicable.

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
