# Peer review of "Pharmacological Activity of Cepharanthine"

_molecules, 2023, doi:10.3390/molecules28135019_

Round 1
Reviewer 1 Report
The work includes a review of the literature and scientific achievements regarding the pharmacological activity of cepharanthine. Cepharantine is a long-known drug with many medicinal properties. This type of review can be used as a compendium of applications of this drug against many diseases. What is also very important recently, inhibitor properties against SARS-CoV-2 have been confirmed. The work has been divided into logical chapters and subchapters. Everything is supported by carefully selected literature.
Two small notes:
1. There is no title in Table 2.
2. If possible, it would be good to insert drawings of better quality. However, I leave that decision to the editors of the journal.
I recommend the work for publication.
Reviewer 2 Report
This manuscript is aimed to review the pharmacological activities, mechanisms, and applications of cepharantine. The scope of the manuscript covers almost all aspects of the biological activities of cepharantine and its applications in a variety of health issues. A few similar reviews have recently also been published e.g. (1) Bailly, Christian. "Cepharanthine: An update of its mode of action, pharmacological properties and medical applications." Phytomedicine 62 (2019): 152956, (2) Rogosnitzky, Moshe, and Rachel Danks. "Therapeutic potential of the biscoclaurine alkaloid, cepharanthine, for a range of clinical conditions." Pharmacological reports 63.2 (2011): 337-347. Nevertheless, this manuscript has the highest coverage and includes the most recent works of literature. Therefore, I believe that this review deserves to be considered for publication in Molecules after revisions according to the comments given below:
1. cepharanthine has been found to be a very effective miracle drug in almost all aspects of the health issues reviewed in this review based on related scientific reports. In my opinion, there should be some discussion on the findings of the cited papers and the potential drawbacks and limitations of the methods used.
2. There are commercial tablets of cepharanthine. There are some side effects reported in the prospectus if there is any study performed on the use of those commercially available cepharanthine tablets. It would be better to place this issue in this review, especially human clinical trials.
3. Although Fig 1A aims to display the inhibition of membrane fusion, it shows the membrane fusion. Figure 1B also shows that the virus entered the cell despite the suggestion of blockage of spike protein with ACE2. Moreover, both A and B figures give the impression that cepharanthine blocks the viral genetic material.
4. Name of most species given have not been written in italics throughout the manuscript, e.g., Lines 230-2261
5. Figures should be placed after where it is cited in the text.
Reviewer 3 Report
The review article by Ke Liu et al. on “Pharmacological Activity of Cepharanthine” is interesting, and I have a strong belief that readers would benefit from it. I have some minor suggestions, please see below;
1. Please put chemical structure of cepharanthine in introduction section.
2. I suggest the authors to add small paragraph describing the mechanism in general way after introduction.
3. In the section of safety and bioavailability of cepharanthine, I did not find any study or discussion about safety. Authors have just mentioned about bioavailability. Please include about safety of this compound. Please include more studies about strategies to improve efficacy of this compound.
4. In addition to their bioactivities, natural products also show side effects and can only be prescribed after clinical validation. Cepharanthine, being a natural product, are there any reports of side effects of Cepharanthine on human cells? Please mention if any, in the section of Safety and bioavailability of cepharanthine.
5. In the section, 2.2. Antitumor activity. Authors have described the potential role of Cepharanthine in cancer management through modulation of different cell signalling pathways. Please put table with dose, cancer cell lines and mechanism of action.
6. It is very important to put sub-heading as conclusion and future prospectives.
Round 2
Reviewer 2 Report
The authors have revised the manuscript in line with my comments. I am satisfied with the corrections and revisions made.